# Research and Application of Microwave Microstrip Transmission Line-Based Icing Detection Methods for Wind Turbine Blades

**DOI:** 10.3390/s25030613

**Published:** 2025-01-21

**Authors:** Min Meng, Xiangyuan Zheng, Zhonghui Wu, Hanyu Hong, Lei Zhang

**Affiliations:** 1Institute for Ocean Engineering, Tsinghua Shenzhen International Graduate School, Shenzhen 518055, China; 13506197273@vip.163.com; 2School of Electrical and Information, Engineering Wuhan Institute of Technology, Wuhan 430205, China; 18389315610@163.com (Z.W.); hhyhong@163.com (H.H.); leizhang0520@163.com (L.Z.)

**Keywords:** microwave, microstrip line, icing thickness, wind turbine

## Abstract

In areas where there is high humidity and freezing rain, there is a tendency of blade icing on wind turbines. It results in energy dissipation and mechanical abrasion and also creates a safety concern due to the risk of having falling ice. Real-time online detection of icing is crucial in the enhancement of power generation efficiency and in the safety of wind turbines. The current methods of icing detection that use ultrasound, optics, vibration, and electromagnetics are already studied. But these methods have their drawbacks, including small detection ranges, low accuracy, large size, and challenges in distributed installation, making it hard to capture the real-time dynamics of the icing and de-icing processes on the wind turbine blades. To this end, this paper presents a new blade surface icing detection technique using microstrip lines. This approach uses the impact of icing state and thickness on the effective dielectric constant of the microstrip line surface. This paper presents the analysis of time-domain features of microwave signals, which facilitates the identification of both the icing state and the corresponding thickness. Simulation and experimental measurement of linear and S-shaped microstrip sensors are used in this research in order to compare the response of the sensors to the variation in the thickness of the icing layer. It is seen that for icing thickness ranging from 0 mm to 6 mm, the imaginary part of the S21 parameter of the S-shaped microstrip line has a more significant change than that of the linear microstrip line. The above experiments also confirm that the phase shift value of the S-shaped microstrip line is always higher than that of the linear microstrip line for the same variation of icing thickness, which proves that the S-shaped microstrip line is more sensitive than the linear one. Also, it was possible to establish the relationship between the phase shift values and icing thickness, which makes it possible to predict the icing thickness. The developed microwave microstrip detection technology is intended for usage in the wind turbine blade icing and similar surface detection areas. This method saves the size and thickness of icing sensors, which makes it possible to conduct measurements at various points. This is especially beneficial for usage in wind turbine blades and can be further applied in aerospace, automotive, and construction, especially the bridges.

## 1. Introduction

Wind energy stands out as one of the most extensively harnessed renewable energy sources at present. Owing to its sustainable supply, environmental benefits, and cost-effectiveness, it is recognized as one of the most promising forms of energy for future development. Public data indicate that from 2017 to 2023, there has been a continuous increase in both the global installed capacity of wind power and the proportion of wind-generated electricity in China’s overall power production, as depicted in Figure 1 and Figure 2.

In recent years, wind power has received people’s attention and extensive development [1]. Onshore wind farms are typically constructed in high-altitude regions, where wind energy potential is substantial [2]. However, during the winter months, the incursion of cold and humid air often leads to significant ice accumulation on wind turbine blades [3,4]. This blade icing presents several critical issues: reduced aerodynamic efficiency, which causes a decline in motor power output and energy production [5,6,7,8,9,10]; diminished motor lifespan due to prolonged overload conditions; and the emergence of safety hazards [11,12,13].

Anti-icing and de-icing systems can effectively mitigate these issues; however, their reliable operation necessitates accurate information from precise icing monitoring systems.

Aligned with the research focus of this paper, the primary methods for icing detection identified in previous studies are summarized in Table 1. These methods include meteorological, visual, capacitive, ultrasonic, infrared, and vibration damping sensors. Meteorological data-driven approaches are indirect measurement techniques that rely on extensive datasets to develop complex models, which often lack high accuracy [14,15]. While other direct measurement methods can provide some information on icing thickness, they are not optimal for real-time monitoring of icing conditions. These methods also face limitations such as small measurement ranges, high algorithmic demands, and susceptibility to environmental factors. Since wind turbine blades operate at high rotational speeds, icing sensors must meet stringent requirements, such as being thin, compact, and lightweight. Currently available icing detection methods fail to satisfy these criteria.

Changes in the relative dielectric constant of different media can be effectively detected using microwave technology. The ice formation process on wind turbine blades, which involves freezing rain striking and freezing upon the blade surface, leading to the accumulation of ice layers through repeated impacts, entails changes in the medium. This makes microwave technology the optimal choice for measuring ice accumulation. As illustrated in Table 2, the microwave icing detection technologies currently in widespread use are primarily categorized into transmission line types and resonant types.

Planar microwave resonators and microstrip patch antennas can measure frost and ice based on the monotonic shift of the resonant frequency. However, because these devices operate at a fixed single resonant frequency, they are susceptible to environmental influences. This typically necessitates compensation or calibration prior to measurement, complicating the attainment of high-precision measurements in harsh conditions. Refs. [24,25] introduce a new microwave planar sensor method, which can effectively improve the sensitivity of liquid concentration detection, but there is still a lack of further research on ice coating detection. In contrast, the transmission line method can function over a broad frequency range and calibrate icing status by monitoring changes in the amplitude or phase of microwave signals. The waveguide method does not offer significant advantages for measuring icing on wind turbine blades, as waveguides cannot fully conform to the blade surface and therefore fail to accurately reflect the actual icing conditions.

In this paper, an innovative approach to the identification of icing conditions is presented by using a microstrip transmission line of small dimensions. Hence, from the phase shift of microwave signals depending on the icing state, it is possible to determine the thickness of the ice. Two microstrip structures will be discussed, the sensitivity of each of these structures to variations in icing thickness will be discussed, and the best of the two will be recommended for installation on wind turbine blades.

Section 2 of this paper seeks to establish the theoretical background of icing measurement with the help of microstrip sensors. The last part is the simulation, verification, and design of the aforementioned sensors. Section 4 describes the system implementation as well as experimentation. Last but not least, Section 5 and Section 6 offers the results and the conclusions of the study.

## 2. Microstrip Transmission Line Detection Theory

The frosting on the blades of wind turbines sticks to the surface of the blades, which in turn affects their performance and output. This section offers a detailed derivation of the concepts that can be used in identifying the thickness of the medium that is on the microstrip line structure. Figure 3 shows a microstrip line when it is plated with icing medium, and the schematic diagram of the structure is shown.

The schematic divides the cross-section of the microstrip line measurement range into three distinct regions: Region ① is the microstrip line dielectric substrate with relative dielectric constant $\varepsilon_{r}$; Region ② is the icing layer, which affects the microstrip line having relative dielectric constant $\varepsilon_{ice}$; and Region ③ is the air with relative dielectric constant $\varepsilon_{air}$. The dimensions *h* and *d* stand for the thickness of the dielectric substrate and icing layer, respectively, and *w* represents the width of the microstrip line. In particular, for the case of a dry air environment, Region ② is absent since the medium above the microstrip line is air. On the other hand, when the temperature and humidity are low, an icing layer occurs on the top side of the microstrip line; hence, all the three regions are present. The difference in the icing thickness *d* on the surface of the microstrip line results in changes in the effective dielectric constant of the microstrip line, and the value depends on the ice thickness.

The effective dielectric constant $\varepsilon_{re}$ under icing conditions can be calculated by the method described in reference [26]. First, it is noted that through the symmetry properties of the potential function of the microstrip line structure and the corresponding boundary conditions, the differential equation for the potential function is solved. This solution gives the variational form of the unit length capacitance when the icing on the microstrip line is of zero thickness. The formula for calculating the capacitance $C_{ice}$ is as follows:(1)$$\frac{1}{C_{ice}}=\frac{1}{2\pi \varepsilon_{0}}\int_{-\infty }^{\infty }\frac{{\left\{\frac{1.6\sin\left(\alpha w/2\right)}{\alpha w/2}+\frac{2.4}{{\left(\alpha w/2\right)}^{2}}\left[\cos\left(\alpha w/2\right)-\frac{2\sin\left(\alpha w/2\right)}{\alpha w/4}+{\left[\frac{\sin\left(\alpha w/4\right)}{\alpha w/4}\right]}^{2}\right]\right\}}^{2}}{\alpha h\left[\varepsilon_{r}\coth\left(\alpha h\right)+\varepsilon_{ice}\frac{\varepsilon_{ice}sh\left(\alpha d\right)+\varepsilon_{air}ch\left(\alpha d\right)}{\varepsilon_{air}sh\left(\alpha d\right)+\varepsilon_{ice}ch\left(\alpha d\right)}\right]}d\left(\alpha h\right)$$

Here, α is the frequency component obtained from the Fourier transform of the potential function and $\varepsilon_{0}$ is the vacuum dielectric constant. To determine the effective dielectric constant $\varepsilon_{re}$, it is necessary to calculate the unit-length capacitance $C_{0}$ of the air microstrip:(2)$$C_{0}=1/cZ_{01}$$

In Equation (2), *c* denotes the speed of light in free space, and the characteristic impedance $Z_{01}$ of the air microstrip can be referenced from the literature [25]:(3)$$Z_{01}=\left\{\begin{array}{cc} 60\ln\left(\frac{8h}{w}+\frac{w}{4h}\right), & \frac{w}{h}<1\\ 120\pi /\left[\frac{w}{h}+1.393+0.667\ln\left(\frac{w}{h}+1.444\right)\right], & \frac{w}{h}\geq 1 \end{array}\right.$$

By integrating the findings from reference [27], the effective dielectric constant $\varepsilon_{re}$ of a microstrip line with surface icing can be expressed as:(4)$$\varepsilon_{re}=C/C_{0}$$

From Equation (3), it can be seen that once the thickness of the microstrip line substrate and the width of the microstrip line have been fixed, $C_{0}$ is constant. Therefore, combining Equation (4), it can be seen that the value of $\varepsilon_{re}$ is directly proportional to the value of $C_{ice}$. To make the further calculation easier, at first, the value of $C_{i0}$ is calculated at the condition when the icing thickness is zero. Next, we find the values of $C_{ix}$ for icing thicknesses of 1 mm to 10 mm in order to examine the effect of the thickness on $\varepsilon_{re}$.(5)$$\frac{\varepsilon_{ren}}{\varepsilon_{re0}}=\frac{C_{ix}}{C_{i0}}$$

Based on Equations (1) and (5), with *w* set to 1.27 mm, *h* set to 1 mm, $\varepsilon_{r}$ set to 4, $\varepsilon_{ice}$ set to 3, and $\varepsilon_{air}$ set to 1, we can calculate the relationship between the icing thickness and the $\varepsilon_{ren}$/$\varepsilon_{re0}$ ratio, as illustrated in Figure 4.

From Figure 4, it is evident that the effective dielectric constant $\varepsilon_{re}$ increases monotonically with the increase in icing thickness, and this trend decelerates as the thickness further increases.

Assuming an ideal microwave signal is transmitted through a microstrip line under icing conditions, the phase shift formula from the starting port to the terminal port is as follows:(6)Δϕ=β∗Δl
where Δϕ is the phase difference between the two ports (rad), β is the phase constant of this section of the microstrip line (rad/m), and Δl is the electrical length of the transmission line (m). The expression for the phase constant is:(7)$$\beta =\frac{2\pi f}{V_{P}}$$
where *f* is the frequency of the microwave signal (Hz), which can be generated by a selected signal generator, and $V_{P}$ is the phase velocity of the microwave signal (m/s), determined by the following expression:(8)$$V_{P}=\frac{c}{\sqrt{\varepsilon_{re}}}$$

Thus, *c* is the speed of light in free space, and $\varepsilon_{re}$ is the effective dielectric constant of the strip transmission line. By combining Equations (6)–(8), the final expression for the phase shift of the microstrip line is [28]:(9)$$\Delta \phi =\frac{2\pi f}{c}*\sqrt{\varepsilon_{re}}*\Delta l$$

From Equation (9), it can be stated that the phase shift value of the microstrip line at a given operating frequency *f* and for a given length of the transmission line Δl is mainly governed by the effective dielectric constant and is a function of its square root. Thus, with the increase in icing thickness *d*, the effective dielectric constant $\varepsilon_{re}$ of the microstrip line also increases and therefore has a higher phase shift value Δϕ. This principle lays the groundwork for the application of microstrip lines in identifying the amount of icing on the blades of wind turbines.

## 3. Simulation and Design of Microstrip Sensors

In this study, we employed High-Frequency Structural Simulator (HFSS) 2021 software to design two distinct structures of microstrip line sensors to evaluate their sensitivity to variations in icing thickness [29]. One sensor features a linear microstrip line, while the other utilizes an S-shaped microstrip line. The schematic dimensions of these sensors are depicted in Figure 5.

As illustrated in Figure 5, the model dimensions for the two sensors were designed to be 62 mm × 32 mm to meet the requirements of practical measurement applications. The HFSS software was utilized to establish an air cavity compatible with the sensors, thereby simulating the actual operational environment. The upper air cavity was set to 15 mm. Rogers RT6010 (Rogers Foam Corporation, USA), a dielectric substrate known for its good isotropy, was selected due to its relative dielectric constant. To ensure that the microwave signal transmission remains consistent and to achieve 50 Ω impedance matching at the ports, the linear microstrip line was designed with a conductor width *Dl* of 1.3 mm and a substrate thickness *Tl* of 0.5 mm. Similarly, the S-shaped microstrip line was designed with a conductor width *Ds* of 1.1 mm and a substrate thickness *Ts* of 0.5 mm.

We typically use the S_21_ reflection parameter to assess the transmission performance of microwave microstrip lines. The S_21_ parameter reflects the extent of microwave signal transmission from port 1 to port 2 of the transmission line, with the real part indicating amplitude variations and the imaginary part indicating phase variations [30]. To ensure optimal transmission characteristics, the sensor’s actual operating frequency was identified within the frequency sweep range of 200 MHz to 1000 MHz. The S_21_ parameter frequency sweep curves for the two microstrip lines are presented in Figure 6.

As depicted in Figure 6, the S_21_ parameters for both microstrip lines diverge from 0 dB as the frequency increases, signifying that higher frequencies diminish the transmission efficiency of the microstrip lines. Taking into account the operating frequency margin, 300 MHz was chosen as the operational frequency for the sensors in this study.

Additionally, icing modules were incorporated into the two microstrip line structures. The thickness variable of the icing module Xice was set within the range of 0 to 6 mm. The schematic representation of the sensors with the icing structure is shown in Figure 7.

Since the imaginary part of the S_21_ parameter represents the phase change during microwave transmission, and Equation (9) has demonstrated that the phase of the transmission line is influenced by the dielectric constant of the icing, the imaginary part of the S_21_ parameter can be used to analyze the phase shift induced by variations in icing thickness. This allows for a comparison of the sensitivity of the two sensors. The simulation results are presented in Figure 8.

From Figure 8, it can be observed that during the icing thickness variation from 0 to 6 mm, the imaginary part of the S_21_ parameter of the S-shaped microstrip line exhibits a more pronounced change compared to that of the linear microstrip line. This indicates that the S-shaped transmission line is more sensitive to changes in icing thickness than the linear transmission line.

## 4. Experimental Procedure

To verify the accuracy of the simulation process and the practical application of icing measurement, PCB designs and physical fabrication of the two microstrip lines were conducted. The physical sensors are shown in Figure 9.

Both microstrip line structures employ back-fed feeding, with the two ports of the microstrip line connected via coaxial connectors. To simulate the actual icing conditions on the surface of wind turbine blades, an icing thickness detection system was constructed, as illustrated in Figure 10. The entire system comprises the following components:

① Measurement sensor module

② Signal triggering and acquisition module

③ Information transmission module

④ Information display and processing module

⑤ Experimental space module

Measurement sensor module: This consists of the microstrip sensors and the measuring box, with a container height of 20 mm, accommodating the icing thickness measurement range of 1 to 6 mm.

Signal triggering and acquisition module: This consists of an MCU controller, a microwave signal generator, and a microwave amplitude and phase detector. The microwave signal generator produces a 300 MHz microwave signal and transmits it to the measurement sensor module. The microwave amplitude and phase detector performs amplitude and phase detection on the microwave signals at both ends of the measurement sensor module, outputting voltage signals related to phase and amplitude changes. Finally, the MCU controller collects the phase and amplitude voltage data, obtaining the relevant phase and amplitude change data.

Information transmission module: This module uses a USB-RS485 (Jiangsu Maihe Internet of Things Technology Co., Ltd., China) serial cable to facilitate data transmission between the control chip and the host computer.

Information display and processing module: This module, based on a Qt application on the host computer, can display and process data in real time.

Experimental space module: This comprises a controllable low-temperature freezer with a temperature range of −25 °C to 0 °C, meeting the requirements for actual scenario simulation.

The specific experimental steps are as follows:

Step 1. Place the sensors in the experimental freezer set to a constant temperature of −15 °C. Monitor the sensors’ temperature using the information display module until they reach the experimental temperature of −15 °C before proceeding with the subsequent steps.

Step 2. Attach a ruler to the side of the measuring box. First, record the phase data at 0 mm icing, then use a pipette to incrementally add low-temperature water solution to the measuring box, recording the phase data at stabilized icing thicknesses from 1 mm to 6 mm.

Step 3. To minimize experimental error, repeat the above process five times and take the average value of the phase data. The experimental procedure for the S-shaped sensor is illustrated in Figure 11.

Similarly, the same procedure is applied to the linear sensor. The experimental steps are illustrated in Figure 12.

## 5. Results and Discussion

In a −15 °C experimental environment, the S-shaped microstrip sensor was employed to measure the phase data for icing thicknesses ranging from 0 to 6 mm. The experiment was repeated five times to compute the average values and standard deviations. The error bars and non-linear fitting of the experimental data are presented in Figure 13.

As illustrated in Figure 13, the ordinate is the voltage signal after amplifying the output phase difference signal of the amplitude detector. Its size reflects the degree to which the phase of the microwave signal changes after passing through the detected transmission line. The phase values of the S-shaped microstrip line are approximately 1260 under pure air conditions. However, when ice forms on the surface of the S-shaped microstrip line, the phase values shift to the range indicated by the shaded area in the figure, spanning from 1343.2 to 1283.6. These phase shifts correspond to ice thicknesses ranging from 1 mm to 6 mm. The non-linear fitting expression for these data points is presented below:(10)$$Phase_{S}=0.24iceh^{5}-4.383iceh^{4}+30.14iceh^{3}-96.82iceh^{2}+151.9iceh+1261$$

Similarly, the same procedure was applied to the linear microstrip sensor. The error bars and the non-linear fitting of the experimental data are depicted in Figure 14.

According to Figure 14, the phase values of the linear microstrip line in pure air conditions are around 816. When the surface is covered with ice, the phase values fall within the shaded region in the figure, ranging from 859.2 to 887.2. These phase shift values correspond to icing thicknesses from 1 mm to 6 mm. The non-linear fitting expression for the data is as follows:(11)$$Phase_{L}=0.1208iceh^{5}-2.162iceh^{4}+14.64iceh^{3}-47.05iceh^{2}+77.37iceh+816$$

A comprehensive analysis, combining Figure 13 and Figure 14, indicates that the S-shaped microstrip sensor exhibits greater variations in phase values during the transition from air to icing, as well as with increasing icing thickness, thereby demonstrating higher sensitivity. This conclusion aligns with the simulation analysis presented in Section 3.

Finally, this paper makes a trial production of an ice sensor, using a solar cell and lithium battery combined power supply and Lora wireless communication. The whole circuit is potted with polyurethane flexible material, the whole has a certain flexibility and tear resistance, and the thickest part is about 5 mm, which can be pasted on the relatively flat part of the wind turbine blade. The surface of the microwave transmission line is exposed to the outside, which facilitates ice covering its surface. The Lora antenna and temperature sensor are pot-sealed in a flexible polyurethane material, which does not affect the transmission of wireless communication signals because the polyurethane material is a non-metallic material. The ice sensor is shown in Figure 15:

In order to reduce power consumption and extend the use time of the lithium battery, the circuit adopts the automatic wake-up mode for 1 to 10 min. Ice thickness and temperature data can be transmitted via Lora wireless communication to ground receiving equipment. When the ice thickness exceeds the set value, you can choose to stop as needed. If the wind turbine is installed with heating and melting equipment, it can be combined with the ice sensor for linkage control. When the ice thickness reaches the set value, the heating and melting will be started. When the ice thickness reaches 0, the heating and melting will be stopped.

## 6. Conclusions

Blade icing on wind turbines is considered one of the primary issues leading to turbine failure. The accumulation of ice on the blade surface reduces aerodynamic efficiency, increases maintenance costs, and poses operational risks. This paper reports on a blade surface icing detection technology based on microstrip lines. The method leverages the impact of different dielectric constants of the media on the surface of the microstrip structure on the effective dielectric constant of the microstrip line, which subsequently alters the phase shift of the microwave signal.

Based on measurement principles and practical applications, both linear and S-shaped microstrip sensors were constructed. During the icing thickness variation from 0 to 6 mm, the imaginary part of the S_21_ parameter of the S-shaped microstrip line exhibited greater changes than that of the linear microstrip line. This indicates that, in numerical simulations, the S-shaped microstrip line is more sensitive to changes in the icing medium thickness compared to the linear microstrip line, validating the feasibility of measuring icing thickness with both sensors.

Moreover, the physical system for icing detection was developed for the experimentation purpose. From −15 °C, when the icing thickness varied from 0 to 6 mm, the phase shift value change of the S-shaped microstrip line was 123.6, which was much larger than the 71.2 of the linear microstrip line. This proves that the S-shaped microstrip line is more sensitive than the straight microstrip line, as proved experimentally. Furthermore, non-linear models were also used in both the sensors to gain an estimate of the icing thickness.

Because the efficiency of the solar power generation panel is not high, and the performance of the lithium battery is poor below −20 ° C, the detection equipment is not very ideal in terms of power supply, and future research will consider the use of perovskite solar panels to improve power generation efficiency.

The microstrip detection technology introduced in this paper is intended for icing detection on wind turbine blades and other surfaces. This method can reach the distributed measuring, can be made according to various material surfaces, and has the advantages of small size and low cost. It can also be used in other areas such as in aviation, road transport, and construction of bridges among others.

## Figures and Tables

**Figure 1 sensors-25-00613-f001:**
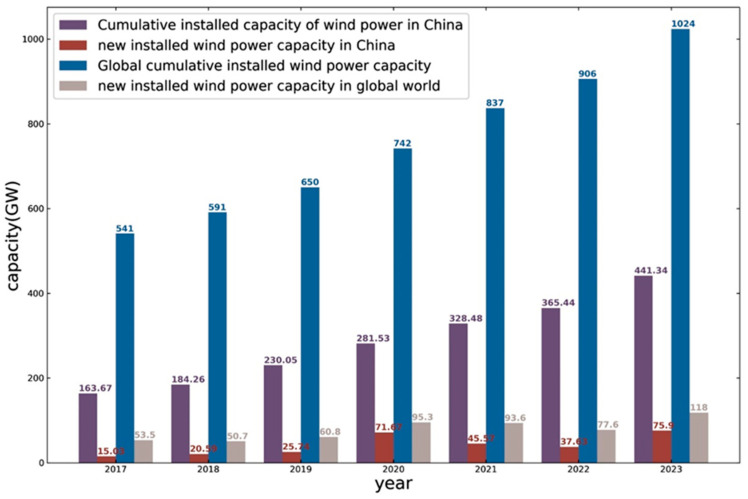
Wind power trends during 2017–2023.

**Figure 2 sensors-25-00613-f002:**
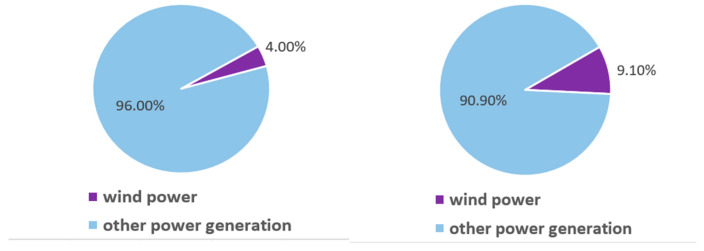
Share of wind power generation in China.

**Figure 3 sensors-25-00613-f003:**
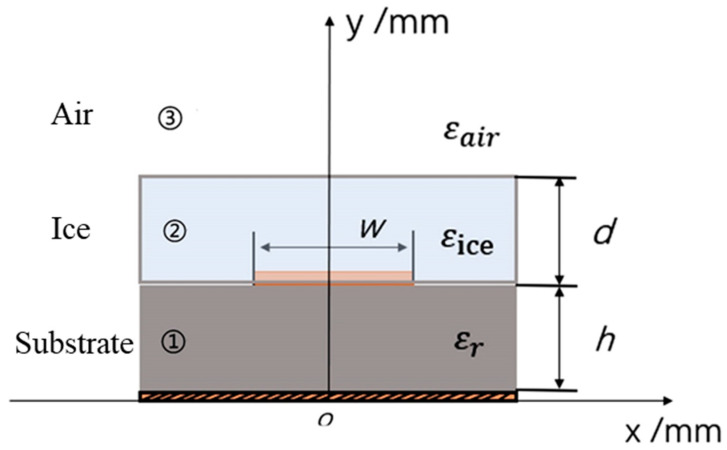
Microstrip line diagram.

**Figure 4 sensors-25-00613-f004:**
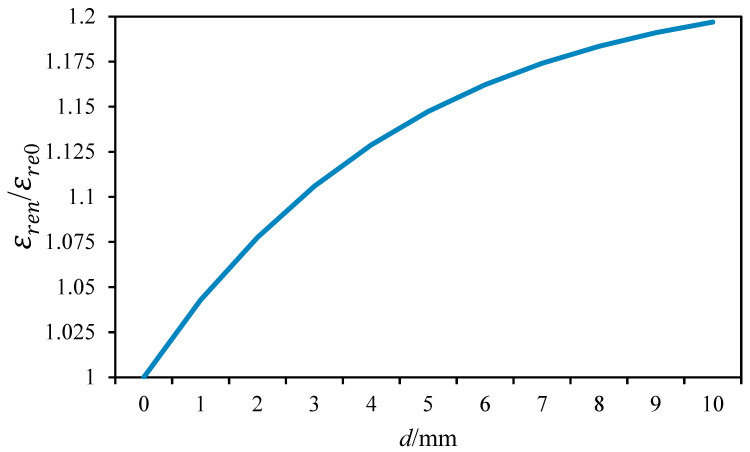
The relationship between ice thickness and the ratio of $\varepsilon_{ren}$/$\varepsilon_{re0}$.

**Figure 5 sensors-25-00613-f005:**
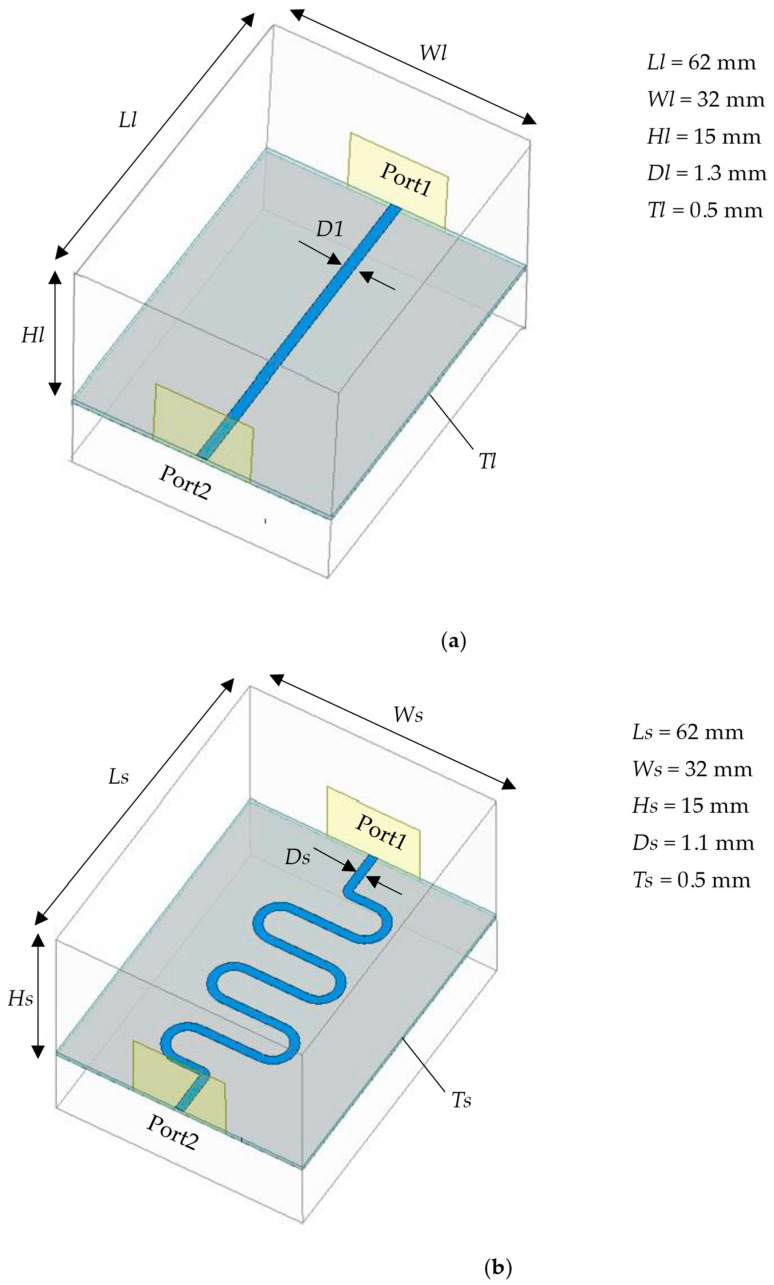
Schematic of the size of two sensors. (**a**) Linear microstrip line; (**b**) S-shaped microstrip line.

**Figure 6 sensors-25-00613-f006:**
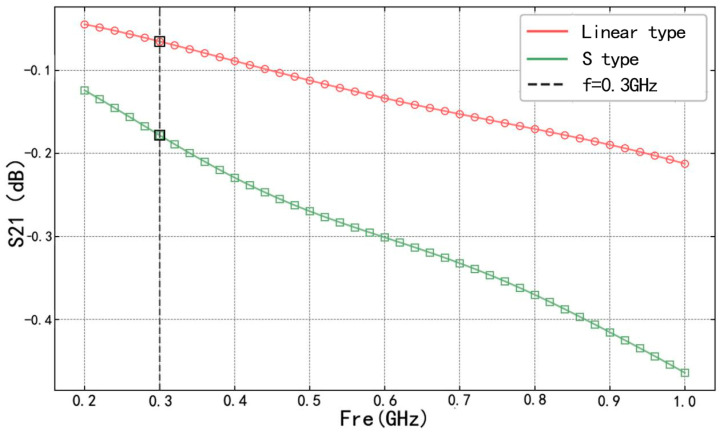
Microstrip line *S*_21_ sweep curve.

**Figure 7 sensors-25-00613-f007:**
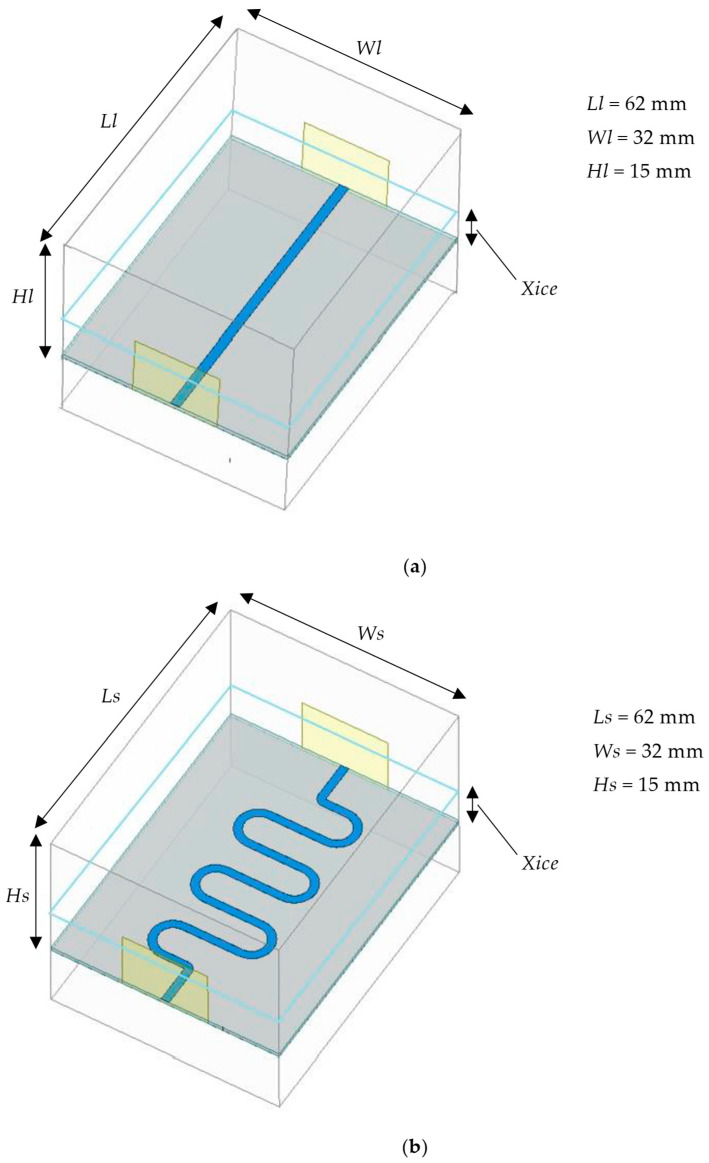
Schematic representation of the sensor’s ice-covered structure. (**a**) Linear microstrip line; (**b**) S-shaped microstrip line.

**Figure 8 sensors-25-00613-f008:**
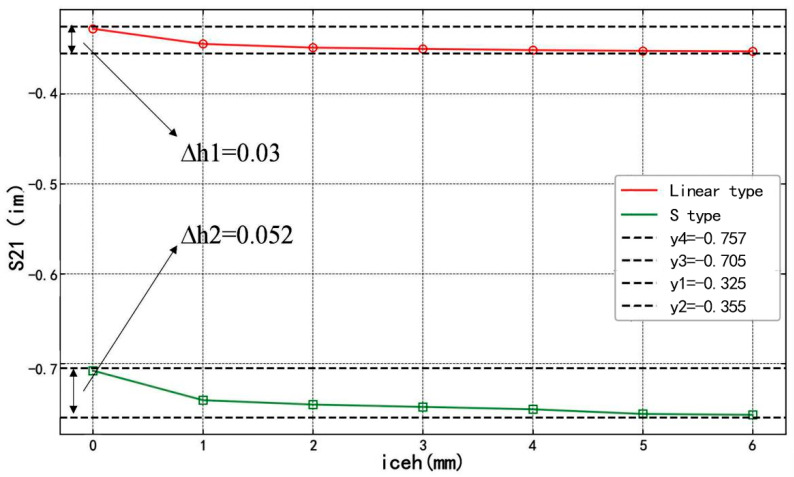
Imaginary part simulation results.

**Figure 9 sensors-25-00613-f009:**
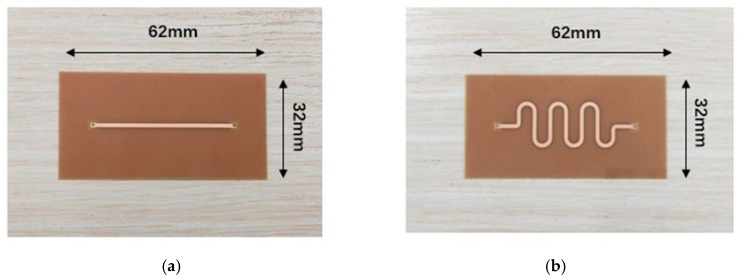
Physical sensors. (**a**) Linear microstrip line; (**b**) S-shaped microstrip line.

**Figure 10 sensors-25-00613-f010:**
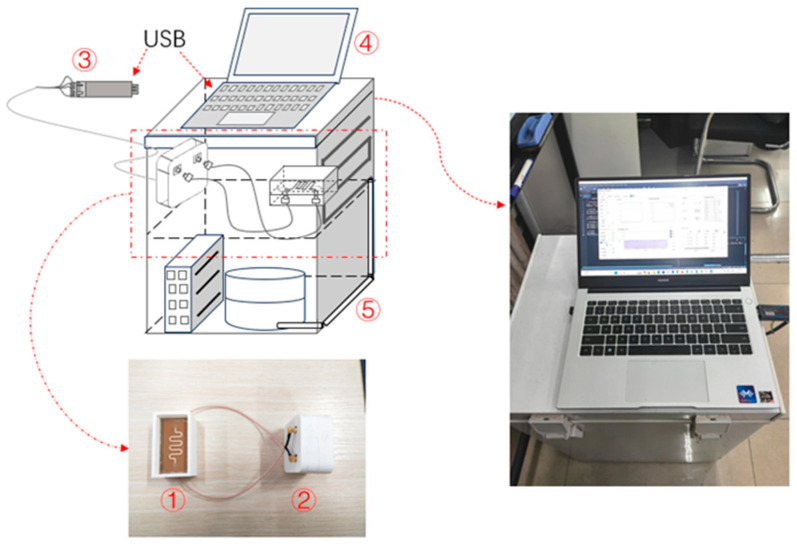
Ice cover thickness detection system.

**Figure 11 sensors-25-00613-f011:**
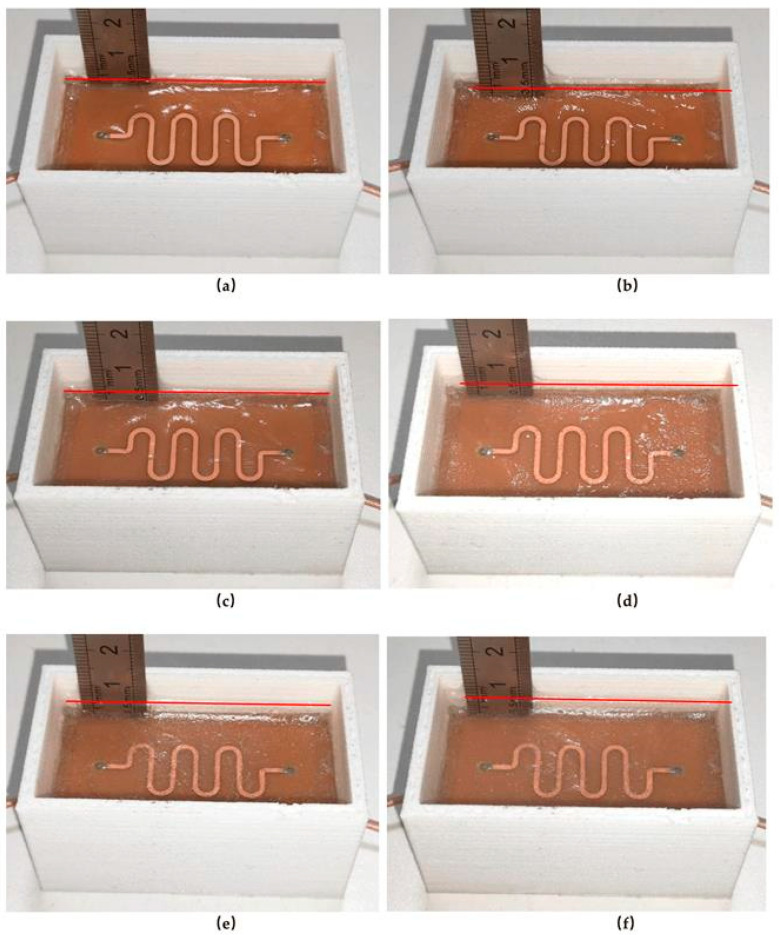
Experimental procedure for S-sensor in terms of ice thickness. (**a**) 1 mm; (**b**) 2 mm; (**c**) 3 mm; (**d**) 4 mm;(**e**) 5 mm; (**f**) 6 mm.

**Figure 12 sensors-25-00613-f012:**
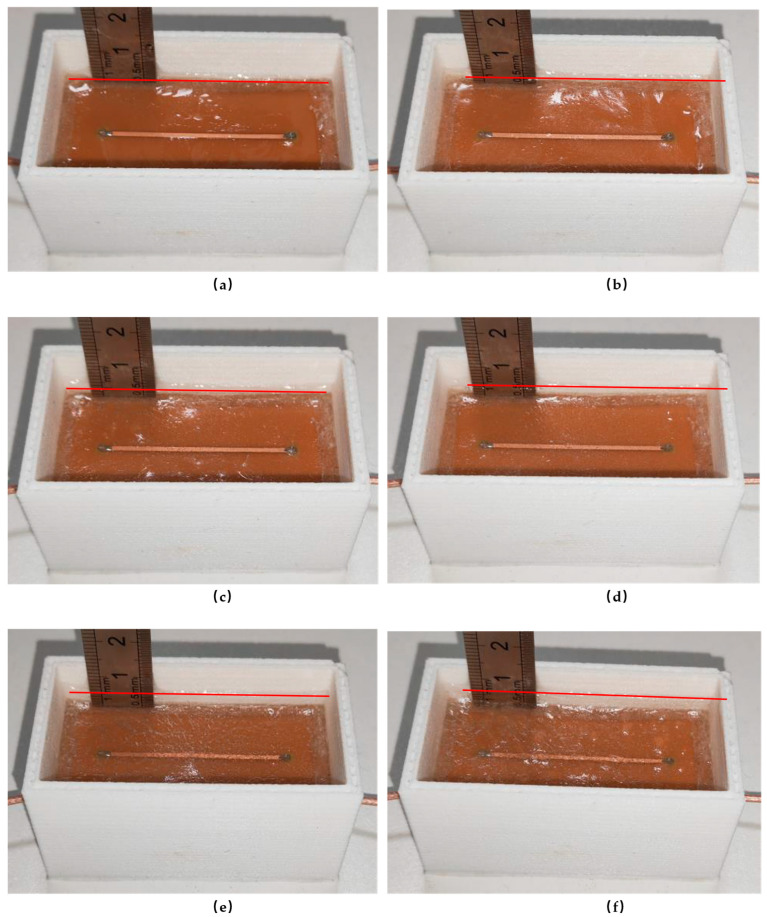
Experimental procedure for linear sensors in terms of ice thickness. (**a**) 1 mm; (**b**) 2 mm; (**c**) 3 mm; (**d**) 4 mm; (**e**) 5 mm; (**f**) 6 mm.

**Figure 13 sensors-25-00613-f013:**
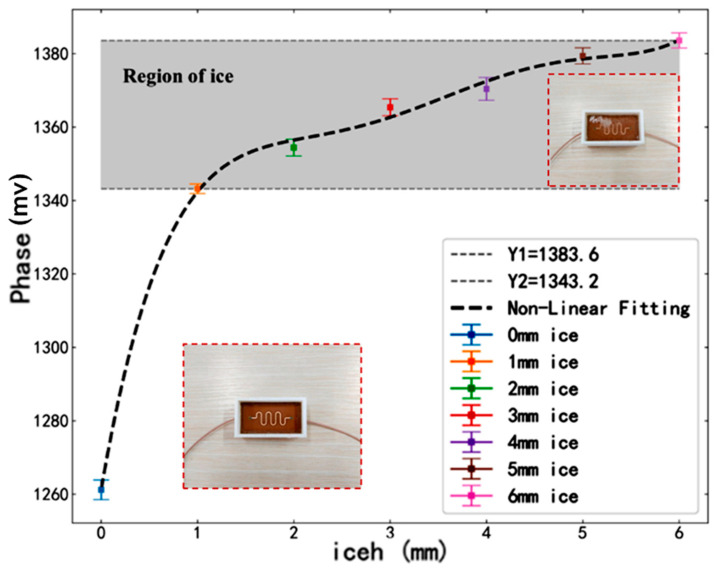
S-shaped microstrip line experimental data error bars and non-linear fitting of data.

**Figure 14 sensors-25-00613-f014:**
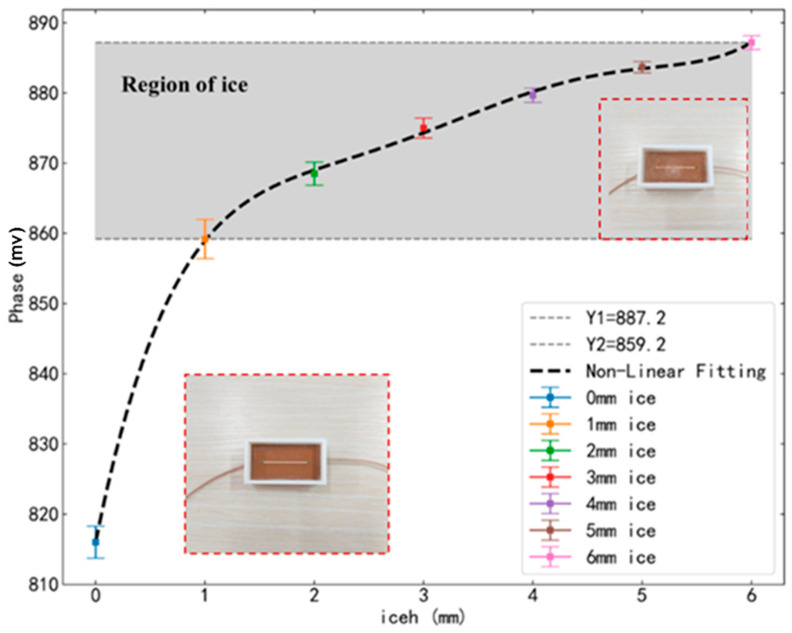
L-shaped microstrip line experimental data error bars and non-linear fitting of data.

**Figure 15 sensors-25-00613-f015:**
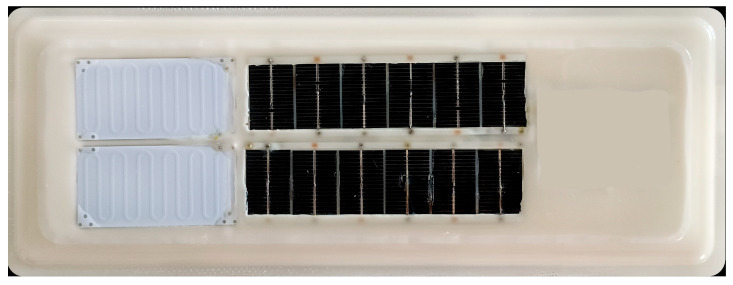
Ice sensor equipment.

**Table 1 sensors-25-00613-t001:** Various ice detection sensors.

Sensor Type	Vantages	Drawbacks	Installation Suitability	Costs
Meteorological sensors and data-driven [14,15]	Real-time data, sensor diversity	High data dependency and complex modeling	Slightly Suitable	High
Visual measurement [16]	Real-time, high-resolution data	Environment-dependent, algorithmically complex	Not Suitable	High
Capacitive [17,18]	Real-time, high sensitivity	Environment-dependent, Small measuring range	Not Suitable	Low
Ultrasonic [19]	Real-time, non-contact	Environment, Materials-dependent	Not Suitable	Low
Infrared [20]	Non-contact, high sensitivity	Thickness limitation, Environment-dependent	Not Suitable	High
Vibration damping [17,18]	Real-time, high sensitivity	Many sources of interference, large error	Slightly Suitable	Low

**Table 2 sensors-25-00613-t002:** Microwave measurement sensors.

Sensor Type	Vantages	Drawbacks	Installation Suitability	Costs
Planar Microwave Resonator [21]	Planar and non-planar measurements	Narrowband, Requires calibration	Slightly Suitable	Low
Microstrip Patch Antenna [22]	Real time, non-contact	Small frequency range, Environment-dependent	Slightly Suitable	Low
Waveguides [23]	Wide operating frequency	Large space volume	Slightly Suitable	Low
Presented Technique	small dimension, Highly flexible, low loss	None	Slightly Suitable	Low

## Data Availability

Data are contained within the article.

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
