# Peer review of "Research and Application of Microwave Microstrip Transmission Line-Based Icing Detection Methods for Wind Turbine Blades"

_sensors, 2025, doi:10.3390/s25030613_

Round 1

Reviewer 1 Report

Comments and Suggestions for Authors

Ice detection is important for many applications. Microwave sensor is of potential application for ice sensing. I have the following comments/questions.

1. I appreciate the author's effort on analytical formulas. However, formulas are usually based on some assumption or simplification. So, how about the accuracy of results presented in Fig.4 if they are compared with numerical simulation?

2. The authors use phase as measurand indicating ice thickness. However, in section 3, the presented simulation results don't include phase results.

3. What's the unit of phase in Fig.13 and 14? I guess it is mV. Furthermore, I am not sure about its physical meaning. Does the phase mean the phase shift of the microstrip line at 300 MHzIt seems not clear that why the authors use phase instead of magnitude for ice thickness sensing.

4. Icing condition may influence permittivity of ice. It may be better if the authors discuss about this. Similarly, potential effect of uniformity of ice thickness can be also mentioned.  

5. Is it possible to install the proposed microwave sensor onto the blade? 

Author Response

Comments 1: I appreciate the author's effort on analytical formulas. However, formulas are usually based on some assumption or simplification. So, how about the accuracy of results presented in Fig.4 if they are compared with numerical simulation?

Response 1: Thank you for your valuable comments. Due to the limitations of the HFSS software we utilized, which lacks the capability to simulate the effective dielectric constant, we were unable to directly simulate the relationship between ice coating thickness and the effective dielectric constant. Instead, we manually defined the ice coating thickness and the dielectric constant of the ice. We proceeded to simulate the variations in ice thickness and the corresponding microwave phase shift. Given that there is no direct formula to derive the relationship between ice coating thickness and microwave phase shift, we first established the correlation between ice coating thickness and effective dielectric constant using relevant formulas, as illustrated in FIG. 4. Subsequently, we employed additional formulas to further derive the relationship between changes in ice coating thickness and microwave phase shift.

Comments 2: The authors use phase as measurand indicating ice thickness. However, in section 3, the presented simulation results don't include phase results.

Response 2: Thank you for your valuable comments. We utilize the S21 reflection parameter to assess the transmission performance of microwave microstrip lines. The S21 parameter indicates the degree of microwave signal transmission from port 1 to port 2 of the transmission line. The real part of the S21 parameter reflects amplitude changes, while the imaginary part corresponds to phase changes. The trend of phase variation with icing thickness is reflected in Figure 8. The thickness of the ice coating has a minor impact on the real part of the S21 parameter but significantly affects the imaginary part; therefore, we employ phase changes to measure the thickness of the ice coating. In practical implementation, we also utilize an amplitude and phase detector to simultaneously monitor changes in both amplitude and phase, and we observe that phase changes are more pronounced than amplitude changes.

Comments 3: What's the unit of phase in Fig.13 and 14? I guess it is mV. Furthermore, I am not sure about its physical meaning. Does the phase mean the phase shift of the microstrip line at 300 MHzIt seems not clear that why the authors use phase instead of magnitude for ice thickness sensing.

Response 3: Thank you for your valuable comments. The unit of phase in FIG. 13 and FIG. 14 is expressed in millivolts (mV), representing the voltage signal amplified by the phase difference output from the amplitude and phase detector. The magnitude of this signal reflects the degree of phase change of the microwave signal after it has traversed the detection transmission line. Our experiments indicate that variations in microwave amplitude do not exhibit sensitivity to ice coating thickness. In contrast, changes in phase demonstrate a greater sensitivity to ice coating thickness. Consequently, we have opted to utilize phase measurements to determine ice thickness. Please refer to the second paragraph of page 14 for specific changes.

“As illustrated in Figure 13, the ordinate is the voltage signal after amplifying the out-put phase difference signal of the amplitude detector. Its size reflects the degree to which the phase of the microwave signal changes after passing through the detected transmission line.”

Comments 4: Icing condition may influence permittivity of ice. It may be better if the authors discuss about this. Similarly, potential effect of uniformity of ice thickness can be also mentioned.

Response 4: Thank you for your valuable comments. Different processes of icing affect the relative permittivity of ice. For example, the permittivity of ice is slightly higher than that of frost. Furthermore, the uniformity of the ice coating impacts the accuracy of ice coating detection. This complexity presents additional challenges and will be the focus of our forthcoming research endeavors.

Comments 5: Is it possible to install the proposed microwave sensor onto the blade?

Response 5: Thank you for your valuable comments. We have successfully implemented the installation of samples on wind power blades, and this setup has been operational for over a year. The thickness of the sensor is approximately 5mm. Below is a photograph of our installation on site.

Reviewer 2 Report

Comments and Suggestions for Authors

see attachment

Author Response

Comments 1: The work is of scientific and practical interest because it explores an important problem: monitoring the condition of wind turbine blades.  These studies make it possible to monitor the operation of such turbines, which should lead to a reduction in emergency situations and an increase in their service life.  The use of S-shaped microstrip lines makes it possible to increase the sensitivity and accuracy of determining the thickness of the ice coating. 

However, the paper does not show how the control of the thick of ice cover is carried out during the operation of the turbine. Therefore, it is advisable to provide a drawing that explains how the thickness of the ice coating is measured during the operation of the turbine.

They offer one of the methods for measuring the thickness of the ice cover, which has some advantages over other methods. One of the methods of icing control of wind turbines is proposed, which can prevent their destruction. If it is wireless, then this is very important for its practical use. As I already wrote in the review, the authors of the work need to provide a drawing that explains how information is read from the sensors when the turbine is running. It is not clear from the work whether this method is wireless or requires connecting cables. This is very important for the practical use of this method in practice

Response 1: Thank you for your valuable comments. We have modified the content, added the introduction of the ice sensor, and analyzed the structure and power consumption. Please refer to the third paragraph of page 15 and the first paragraph of page 16 for details.

Finally, the paper makes a trial production of ice sensor, using solar cell and lithium battery combined power supply, Lora wireless communication. The whole circuit is potted with polyurethane flexible material, the whole has a certain flexibility and tear resistance, and the thickest part is about 5mm, which can be pasted on the relatively flat part of the wind turbine blade. The surface of the microwave transmission line is exposed to the out-side, which facilitates ice covering its surface. The Lora antenna and temperature sensor are pot-sealed in a flexible polyurethane material, which does not affect the transmission of wireless communication signals because the polyurethane material is a non-metallic material. The ice sensor is shown in Figure 15:

Figure 15. Ice sensor equipment.

       In order to reduce power consumption and extend the use time of the lithium battery, the circuit adopts the automatic wake-up mode for 1 to 10 minutes. Ice thickness and temperature data can be transmitted via Lora wireless communication to ground receiv-ing equipment. When the ice thickness exceeds the set value, you can choose to stop as needed. If the wind turbine is installed with heating and melting equipment, it can be combined with the ice sensor for linkage control. When the ice thickness reaches the set value, the heating and melting will be started. When the ice thickness reaches 0, the heat-ing and melting will be stopped.

Reviewer 3 Report

Comments and Suggestions for Authors

Dear authors,

The idea presented in this manuscript shows promise and demonstrates potential. The writing is clear, with measurements aligning well with simulations and yielding good results. However, several aspects need further clarification:

  1. What is the reason for the choice of frequency?
  2. It would be more helpful to have a picture of the back of sensors in Fig. 9 for visibility.
  3. The transmission phase of the structure might be influenced by surrounding proximity variations or slight variation in the location of cable. What do authors propose to minimize these risks?
  4. Comparison Table II - Needs to be improved with considering the following recent relevant sensor methodologies and compare it with others in the table:
    • DOI: 10.1109/TCSI.2020.3003010
    • DOI: 10.3390/s23136236
  5. Discussion of Limitations and Future Work: A discussion section outlining the limitations of the proposed system and potential avenues for future research would be beneficial. Addressing challenges such as scalability, robustness, and power consumption could guide future efforts in improving the system's performance and applicability in real-world scenarios.

Author Response

Comments 1: What is the reason for the choice of frequency?

Response 1: Thank you for your valuable comments. The purpose of selecting the frequency is to better adapt to the detection transmission line and the amplitude and phase detector circuit. When the length of the transmission line is fixed, an increase in frequency results in a shorter microwave wavelength, which in turn leads to a larger phase shift on the fixed-length transmission line and enhances detection sensitivity. However, the phase detector has a certain phase detection range, for example, we use a 180 degree range of phase detector, if the microwave phase shift from no ice to maximum ice conditions change by more than 180 degrees, then it can not be effectively detected. Therefore, the appropriate frequency is selected in order to simultaneously improve the sensitivity and not exceed the phase discriminator range.

Comments 2: It would be more helpful to have a picture of the back of sensors in Fig. 9 for visibility.

Response 2: Thank you for your valuable comments. We took a picture of the back of the sensor as shown in the picture below:

Comments 3: The transmission phase of the structure might be influenced by surrounding proximity variations or slight variation in the location of cable. What do authors propose to minimize these risks?

Response 3: Thank you for your valuable comments. In response to the problem of the surrounding environment and the influence of micro deformation, we take two measures. The first is that we used flexible materials for potting and fixed the circuit board and coaxial transmission line to reduce the deformation effect of the sensor itself.  The second is that we have adopted some algorithms in practical applications.  For example, when the ambient temperature is higher than 0 degrees and it is ensured that the sensor is not covered with ice, we use this opportunity to automatically calibrate the ice thickness to 0, thus eliminating certain Effects of environment and deformation.  We can treat this effect as a systematic error and eliminate it algorithmically.

Comments 4: Comparison Table II - Needs to be improved with considering the following recent relevant sensor methodologies and compare it with others in the table:

     DOI: 10.1109/TCSI.2020.3003010

     DOI: 10.3390/s23136236

Response 4: Thank you for your valuable comments. After reading these two articles, we cited these two journals in the article. For details, please see the first paragraph on page 4.

[24] and [25] introduce a new microwave planar sensor method, which can effectively improve the sensitivity of liquid concentration detection, but there is still a lack of further research on ice coating detection.

Comments 5: Discussion of Limitations and Future Work: A discussion section outlining the limitations of the proposed system and potential avenues for future research would be beneficial. Addressing challenges such as scalability, robustness, and power consumption could guide future efforts in improving the system's performance and applicability in real-world scenarios

Response 5: Thank you for your valuable comments. We have added some discussion on the limitations of the product, explained the flexibility and so on in paragraph 1 on page 16, and expounded some shortcomings of the current power supply and prospects in paragraph 5 on page 16.

“In order to reduce power consumption and extend the use time of the lithium battery, the circuit adopts the automatic wake-up mode for 1 to 10 minutes. Ice thickness and temper-ature data can be transmitted via Lora wireless communication to ground receiving equipment. When the ice thickness exceeds the set value, you can choose to stop as needed. If the wind turbine is installed with heating and melting equipment, it can be combined with the ice sensor for linkage control. When the ice thickness reaches the set value, the heating and melting will be started. When the ice thickness reaches 0, the heating and melting will be stopped.

Because the efficiency of the solar power generation panel is not high, and the per-formance of the lithium battery is poor below -20 ° C, the detection equipment is not very ideal in terms of power supply, and the future will consider the use of perovskite solar panels to improve power generation efficiency.”

Round 2

Reviewer 1 Report

Comments and Suggestions for Authors

I appreciate the author's effort on the improvement of the manuscript. All of my comments/questions are clearly explained.

Reviewer 3 Report

Comments and Suggestions for Authors

Thanks for applying comments